

# Identification of evolutionary relationships and DNA markers in the medicinally important genus *Fritillaria* based on chloroplast genomics

Tian Zhang[1], Sipei Huang[1], Simin Song[1], Meng Zou[1], Tiechui Yang[2], Weiwei Wang[1], Jiayu Zhou[1] and Hai Liao[1]

[1] School of Life Science and Engineering, Southwest Jiaotong University, Chengdu, Sichuan, China
[2] Qinghai lvkang Biological Development Co., Ltd, Xining, Qinghai, China

Corresponding authors
Jiayu Zhou,
spinezhou@home.swjtu.edu.cn
Hai Liao,
ddliaohai@home.swjtu.edu.cn

## ABSTRACT

The genus *Fritillaria* has attracted great attention because of its medicinal and ornamental values. At least three reasons, including the accurate discrimination between various *Fritillaria* species, protection and sustainable development of rare *Fritillaria* resources as well as understanding of relationship of some perplexing species, have prompted phylogenetic analyses and development of molecular markers for *Fritillaria* species. Here we determined the complete chloroplast (CP) genomes for *F. unibracteata*, *F. przewalskii*, *F. delavayi*, and *F. sinica* through Illumina sequencing, followed by *de novo* assembly. The lengths of the genomes ranged from 151,076 in *F. unibracteata* to 152,043 in *F. przewalskii*. Those CP genomes displayed a typical quadripartite structure, all including a pair of inverted repeats (26,078 to 26,355 bp) separated by the large single-copy (81,383 to 81,804 bp) and small single-copy (17,537 to 17,569 bp) regions. *Fritillaria przewalskii*, *F. delavayi*, and *F. sinica* equivalently encoded 133 unique genes consisting of 38 transfer RNA genes, eight ribosomal RNA genes, and 87 protein coding genes, whereas *F. unibracteata* contained 132 unique genes due to absence of the *rps16* gene. Subsequently, comparative analysis of the complete CP genomes revealed that *ycf1*, *trnL*, *trnF*, *ndhD*, *trnN-trnR*, *trnE-trnT*, *trnN*, *psbM-trnD*, *atpI*, and *rps19* to be useful molecular markers in taxonomic studies owning to their interspecies variations. Based on the comprehensive CP genome data collected from 53 species in *Fritillaria* and *Lilium* genera, a phylogenomic study was carried out with three *Cardiocrinum* species and five *Amana* species as outgroups. The results of the phylogenetic analysis showed that *Fritillaria* was a sister to *Lilium*, and the interspecies relationships within subgenus *Fritillaria* were well resolved. Furthermore, phylogenetic analysis based on the CP genome was proved to be a promising method in selecting potential novel medicinal resources to substitute current medicinal species that are on the verge of extinction.

## INTRODUCTION

The genus *Fritillaria* (*Liliaceae*), consisting of 140 known species, is widely distributed in Europe, Asia, and North America (*Huang et al., 2018*; *Rix, 2001*). Based on the Flora of China, twenty-two species are distributed throughout most provinces in China, among which four are diversity hotspots (Xinjiang Plain, East China Plain, Hengduan Mountains, and Northeast Plain). *Fritillaria* species have attracted much attention because they are widely used in traditional Chinese medicine and sometimes as ornamental plants. Dried bulbs from 11 species used singly or as components in traditional Chinese medicine preparations are recorded on Chinese Pharmacopeia (2020), and they are divided into five main concoctions, including Chuan-Bei-Mu (bulbs of the complex group of *F. cirrhosa*), Yi-Bei-Mu (bulbs of *F. palilidiflora* and *F. walujewii*), Zhe-Bei-Mu (bulbs of *F. thunbergii*), Ping-Bei-Mu (bulbs of *F. ussuriensis*), and Hubei-Bei-Mu (bulbs of *F. hupehensis*). Although the bulb of each original species has its own unique curative effect and bioactive compounds and should be used separately for given purposes in traditional prescription, various *Fritillaria* species are still used indiscriminately in clinical prescription due to their similar morphology and names. Particularly, the morphological traits in the group that includes *F. cirrhosa*, *F. unibracteata*, *F. przewalskii*, *F. delavayi*, *F. taipaiensis*, and *F. wabuensis*, are extremely complex due to several highly variable characteristics including stem length, petal color, capsule winged or not, leaf curling, and scale number (*Luo & Chen, 1996*). Therefore, it is vital to carry out taxonomic identification of various *Fritillaria* species.

Molecular systematics has been widely used to clarify angiosperm phylogeny (*Yang et al., 2016*). Firstly, accurate identification (*e.g.*, using DNA markers) has been necessary to discriminate between the *Fritillaria* species and its adulterants. Secondly, since the bulbs of some *Fritillaria* species show great economic values in Asian countries (*Yeum et al., 2007*) and have long been used in traditional Chinese medicine, the wild *Fritillaria* populations have experienced a sharp decline due to long-term overharvesting. To date, four species of Chuan-Bei-Mu and eight species in Xinjiang Plain have been classified as rare resources based on the list of rare endangered higher plants in China (*Li et al., 2018*). DNA markers have been helpful in understanding accurately the genetic diversity and structure of *Fritillaria* population, and thus can be an effective scientific approach for conservation purpose. Thirdly, a better understanding of the relationships within the genus could be of great significance for the medicinal use of *Fritillaria*. Some of the phylogenetically close species might be analyzed for their potential medicinal values to determine if they can be used as substitutes for species that are currently rare. Finally, the phylogenetic positions of some medicinal *Fritillaria* species, such as *F. pallidiflora*, *F. wabuensis*, and *F. davidii*, remain elusive. *Fritillaria pallidiflora* has always been considered a member of the subgenus *Fritillaria* by *Rix (2001)*, whereas *Rønsted et al. (2005)* linked it to the subgenus *Petillium* based on the results of molecular and morphological analyses. *Fritillaria wabuensis* was firstly discovered and nominated as a new species in *Fritillaria* (*Tang & Yue, 1983*), but later it was classified as a variant of *F. crassicaulis* (*Luo & Chen, 1996*) and *F. unibracteata* (*Liu, Wang & Chen, 2009*),

respectively. Therefore, well resolved molecular phylogenies of *Fritillaria*, especially the medicinal species, are necessary.

Currently, the genus *Fritillaria* is divided into eight subgenera, including *Liliorhiza* (including species mainly in North America), *Japonica* (including species mainly in Japan), *Fritillaria* (the biggest subgenus), *Rhinopetalum*, *Petilium*, the monotypic *Davidii* (including only *F. davidii*), *Theresia* (only *F. persica*), and *Korolkowia* (only *F. sewerezowi*), by *Rix (2001)*. At present, despite the frequent usage of nuclear DNA internal transcribed spacer (ITS) and several plastid genome regions (*trnL-trnF*, *matK*, *rbcL*, and *rpl16*) in the classification of this genus, previous studies have found that these markers only provided weak phylogenetic signal. *Rønsted et al. (2005)*, who contributed to the current understanding of evolutionary relationships within *Fritillaria*, investigated the phylogenetic position of 37 *Fritillaria* species in detail using *matK*, *rpl16* intron, and ITS. Consequently, *Fritillaria* was shown to be of two clades, with one clade mainly including species from the North American subgenus *Liliorhiza* and the other clade from the seven remaining subgenera. Consistent with the result of *Rønsted et al. (2005)*, *Khourang et al. (2014)* revealed that the subgenus *Fritillaria* was a sister to the subgenus *Rhinopetalum* based on the phylogenetic tree constructed using the ITS and *trnL-trnF* regions of nine Iranian species. However, *Day et al. (2014)* showed that the largest subgenus (subgenus *Fritillaria*) appeared to be polyphyletic and formed two clades using *matK* and *rbcL* sequences, with one clade comprising taxa occurring mainly in Europe, the Middle East, Japan, and North Africa, and the other clade comprised taxa distributing in China and Central Asia. In our previous research, various *Fritillaria* species from China were classified as North China group and South China group based on the ITS2 sequences, but 57.1% of those species were not effectively resolved (*Zheng et al., 2019*). Recently, *Li et al. (2014)* presented high-quality chloroplast genome using single molecule real-time sequencing, and suggested that *rps19* gene varied the greatest among various species. However, the noncoding regions showed higher variability and were potential effective molecular markers. Therefore, it was proposed that genomics based on the entire chloroplast genome sequences might help identify molecular markers with higher resolution (*Xue et al., 2019*).

The chloroplast (CP) genome has been extensively used for understanding phylogenetic relationships and discovering more effective molecular markers, some of which, such as *trnH-psbA*, *matK*, and *rpl16*, have been used as universal plant DNA barcodes (*Bansal et al., 2018*; *Vinnersten & Bremer, 2001*). To date, there are 23 *Fritillaria* CP genomes that are available in GenBank and they can be used to enhance our understanding of the phylogenetic relationships and to identify molecular markers. Although previous reports (*Bi et al., 2018*; *Chen, Wu & Zhang, 2019*; *Chen, Wu & Zhang, 2020*; *Huang et al., 2020*; *Park et al., 2017*) performed comparative analyses with *Fritillaria* CP genomes available in GenBank, but species-specific identification has not been reported and the phylogenetic place of some ambiguous species remains elusive. At the initial stage of our study, the CP genomes of three important medicinal species (*F. unibracteata*, *F. przewalskii*, and *F. delavayi*) and *F. sinica* had not been reported before 2018. The increasing CP genomes may not only provide a better phylogenetic analysis of this

genus, but also promisingly promote the development of species-specific identification method in the future. Therefore, the CP genomes of these *Fritillaria* species were determined using the Illumina platform in the present study. The objectives of this study included (1) analyzing the global structural patterns of the four CP genomes and comparing them with the available 23 CP genomes of *Fritillaria*; (2) assessing the phylogenetic relationships of the 11 medicinal species used in traditional Chinese medicine, so as to understand the phylogenetic position of some ambiguous species and find potential medicinal plants; and (3) obtaining candidate DNA markers (repeat sequences, SSRs, divergent regions, and indels).

## MATERIALS AND METHODS

### Plant material

The fresh leaves of *F. unibracteata*, *F. przewalskii*, *F. delavayi* and *F. sinica* were collected from the Huzhu County (36°50′15″N, 101°57′06″E), Xining City, Qinghai Province, respectively. The Huzhu County is located in north Hengduan Mountains and east of the Qinghai-Tibetan Plateau. All samples were immediately frozen in liquid nitrogen and stored at −80 °C until DNA extraction.

### Chloroplast genome sequencing and assembly

Total genomic DNA was isolated from 100 mg of fresh leaves using a modified CTAB method. The DNA concentration (>50 ng μL$^{-1}$) was measured using a NanoDrop spectrophotometer. The isolated DNA was fragmented into small pieces using sonication. After end reparation and A-tailing, the short DNA fragments were ligated with the Illumina paired-end adaptors. Based on gel electrophoresis, the suitable fragments were purified and selected as templates for next-step PCR amplification to create the final DNA library. The quality and quantity of the DNA library were measured using the Agilent 2100 Bioanalyzer. Finally, the library was sequenced from both the 5′ and 3′ ends using Illumina NovaSeq6000 PE150 Sequencing platform (Illumina, CA, USA). By use of NGSQCToolkit v2.3.3, the raw reads were filtered to remove the linker sequence and low-quality reads defined as having more than 10% bases with Q-value <20, and thus high-quality clean reads were obtained. The clean reads were then assembled using SPAdes (*Bankevich et al., 2012*) 3.10.1 (http://cab.spbu.ru/software/spades/) software with CP genome of *F. cirrhosa* as reference (NCBI accession number NC_024728.1). Finally, LSC/IR and SSC/IR junctions were further verified by Sanger sequencing.

### Genome annotation and sequence alignment

In order to predict putative gene function, the CDS, rRNA and tRNA genes were aligned using blast v2.2.25 (https://blast.ncbi.nlm.nih.gov/Blast.cgi), HMMER v3.1b2 (http://www.hmmer.org/) and aragorn v1.2.38 (http://130.235.244.92/ARAGORN/), respectively, with *E*-value of 10$^{-5}$. The OGDRAW (https://chlorobox.mpimp-golm.mpg.de/OGDraw.html) helped to make the CP genome maps of *F. unibracteata*, *F. przewalskii*, *F. delavayi*, and *F. sinica*.
The vmatch v2.3.0 (http://www.vmatch.de/) could identify their scattered repetitive sequences (*Askitis & Sinha, 2010*). MISA v1.0 (MIcroSAtellite identification tool, http://pgrc.ipk-gatersleben.de/misa/misa.html) helped to analyze CPSSR. The mafft v7.310 was used to perform indel identification (*Katoh & Standley, 2013*). After using the mafft to align the CP genome sequences, BioEdit software was used to adjust the sequences manually (*Gupta et al., 2014*). DanSP v6.0 was used to perform sliding window analysis (step size = 200 bp and window length = 600 bp) for nucleotide variability (pi) in the whole CP genome (*Rozas et al., 2017*).

## Phylogenetic analysis

The phylogenetic analysis was firstly performed based on *matK*, *psbA-trnH* and *rpl16*, respectively, by use of neighbor-joining (NJ) and maximum-likelihood (ML) methods. Then, the CP genomes in the phylogenetic analysis included the 27 *Fritillaria* species, 26 *Lilium* species, three *Cardiocrinum* species and five *Amana* species. The CP genome evolutionary tree was constructed by BLAST2OGMSA script (https://github.com/fenghen360/BLAST2OGMSA) (*Bi, 2018*) and MEGA-X software (*Kumar et al., 2018*). Firstly, multi-sequence alignment was conducted using BLAST tool of NCBI (*Johnson et al., 2008*), and then the initial alignment result was extracted by BLAST2OGMSA script to obtain homologous blocks. It was reported that BLAST2OGMSA relied on progressiveMauve, a kind of anchored alignment algorithm, to determine where locally collinear blocks (LCBs) represented the landmarks among organelle genomes (such as chloroplast and mitochondrial genomes). The co-exist LCBs among all organelle genomes were extracted and prepared for the further phylogenetic tree construction. In this study, the conserved CDS genes, functional non-coding regions, and rRNA genes as well were combined by BLAST2OGMSA. Finally, the alignment data from BLAST2OGMSA was imported into MEGA-X software to construct the phylogenetic tree using the NJ and ML methods, respectively.

# RESULTS

## Genome sequencing, assembly, and genome features

Based on a stringent quality control, a total of 23,755,399 to 26,831,529 paired-end reads were obtained, generating 7,126,619,700 to 8,049,458,700 clean bases data, from the four *Fritillaria* species. The resultant clean paired-end reads were then employed to assemble the CP genome using the complete genome sequence of *F. cirrhosa* as the reference. Totally, 471,385 to 652,632 mapping reads yielded an average coverage of 934X to 1292X for each species, generating four near full-length CP genomes that ranged from 151,076 in *F. unibracteata* to 152,043 in *F. przewalskii*. The CP genomes contained identical structures, such as two IR regions (26,078 to 26,355 bp each) that were separated by a LSC region (81,383 to 81,804 bp) and a SSC region (17,537 to 17,569 bp) (Fig. 1 and Table S1).

A total of 133 genes were annotated, including 87 protein-coding genes (PCGs), 38 tRNA, and 8 rRNA genes. The global gene order and content were identical in the four species, except that *F. unibracteata* was absent of the *rps16* gene. 21 genes were duplicated
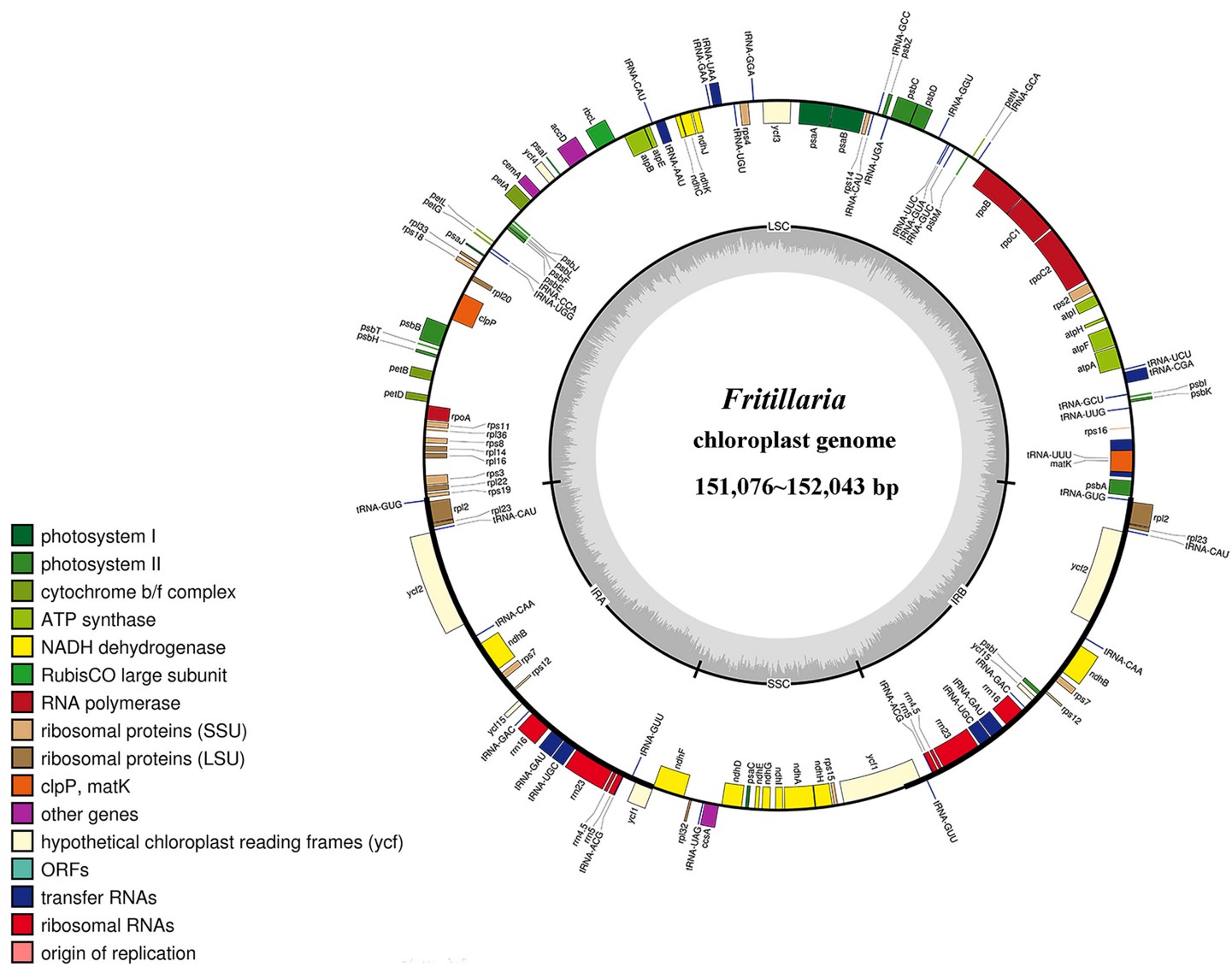

**Figure 1 Chloroplast genome maps of *F. unibracteata*, *F. przewalskii*, *F. delavayi* and *F. sinica*.** Chloroplast genome maps of *F. unibracteata*, *F. przewalskii*, *F. delavayi* and *F. sinica*. Genes belonging to functional group are color-coded. The positive coding gene is located on the outside of the circle, and the reverse coding gene is located on the inside of the circle. The grey circle inside circle represents the GC content.

in the CP genome, including eight tRNA genes, four rRNA genes, and nine PCGs. There were 13 genes containing introns, among which *clpP* and *ycf3* each had two introns, whereas the other 13 genes each had one intron. Eight, one, four, and two introns were located in the LSC, SSC, IRa, and IRb region, respectively (Table 1 and Table S2). Table S2 listed the 15 intron-containing genes in the CP genome of *F. unibracteata*, and those of *F. przewalskii*, *F. delavayi*, and *F. sinica* were included in Tables S3–S5, respectively.

Four *Fritillaria* species showed high sequence similarity (>90% identity). IR regions showed a lower level of sequence divergence than LSC and SSC regions. Contraction and expansion of IR regions, especially the boundary region, are important aspects of CP genomes, which are the main reason of length variation in these genomes (*Abdullah et al.*,

**Table 1 List of annotated genes in four CP genomes.**

| Category | Group of gene | Name of gene |
|---|---|---|
| Photosynthetic | Subunits of photosystem I | *psaA, psaB, psaC, psaI, psaJ* |
| | Subunits of photosystem II | *psbA, psbB, psbC, psbD, psbE, psbF, psbH, psbI (*2), psbJ, psbK, psbL, psbM, psbT, psbZ* |
| | Subunits of NADH dehydrogenase | *ndhA, ndhB (*2), ndhC, ndhD, ndhE, ndhF, ndhG, ndhH, ndhI, ndhJ, ndhK* |
| | Subunits of cytochrome b/f complex | *petA, petB, petD, petG, petL, petN* |
| | Subunits of ATP synthase | *atpA, atpB, atpE, atpF, atpH, atpI* |
| | Large subunit of rubisco | *rbcL* |
| Self-replication | Proteins of large ribosomal subunit | *rpl2(*2), rpl14, rpl16, rpl20, rpl22, rpl23(*2), rpl32, rpl33, rpl36* |
| | Proteins of small ribosomal subunit | *rps2, rps3, rps4, rps7(*2), rps8, rps11, rps12(*2), rps14, rps15, rps16*, rps18, rps19* |
| | Subunits of RNA polymerase | *rpoA, rpoB, rpoC1, rpoC2* |
| | Ribosomal RNAs | *rrn23s (*2), rrn16s (*2), rrn5s (*2), rrn4.5s (*2),* |
| | Transfer RNAs | *tRNA-UUU, tRNA-UUG, tRNA-UUC, tRNA-UGU, tRNA-UGG, tRNA-UGC (*2), tRNA-UGA, tRNA-UCU, tRNA-UAG, tRNA-UAA, tRNA-GUU (*2), tRNA-GUG (*2), tRNA-GUC, tRNA-GUA, tRNA-GGU, tRNA-GGA, tRNA-GCU, tRNA-GCC, tRNA-GCA, tRNA-GAU (*2), tRNA-GAC (*2), tRNA-GAA, tRNA-CGA, tRNA-CCA, tRNA-CAU (*4), tRNA-CAA (*2), tRNA-ACG (*2), tRNA-AAU* |
| Biosynthesis | Maturase | *matK* |
| | Protease | *clpP* |
| | Envelope membrane protein | *cemA* |
| | Acetyl-CoA carboxylase | *accD* |
| | c-type cytochrome synthesis gene | *ccsa* |
| Unknown function | Conserved hypothetical chloroplast | *ycf1 (*2), ycf2 (*2), ycf3, ycf4, ycf15 (*2)* |

**Note:**
(*2) indicated genes duplicated in the CP genomes.

*2020*). As shown in Fig. 2, these 12 *Fritillaria* species had the same gene contents and arrays in IR regions that were expanded in *rps19* and *ycf1* genes. The *rps19* gene in the 12 *Fritillaria* species crossed the LSC/IRb boundary and showed the same length of 279 bp which was similar to that of *Lilium superbum*, except that *F. cirrhosa* had *rps19* gene of 285 bp. In the LSC region, the length of *rps19* gene ranged from 250 to 268 bp, whereas that of *rps19* gene in the IRb region varied from 11 to 35 bp. Besides, the *rps19* genes lost their protein-coding function due to incomplete gene duplication. The similar event was also observed in the *ycf1* gene at the IRb/SSC border. The *ycf1* gene was largely located in the IRb and extended 16 to 32 bp into the SSC region, whereas the *ycf1* gene in *F. taipaiensis* was fully located in the IRb region, 58 bp from the IRb/SSC boundary. In the SSC/IRa boundary of 12 *Fritillaria* species, *ycf1* was a key gene and almost equally distributed. *Ycf1* gene had an SSC region of 4,320 bp in *F. unibracteata* and *F. przewalskii*, but 4,314 bp in *F. delavayi*, *F. sinica*, *F. cirrhosa*, and *F. taipaiensis*, and also had an IRa region of 1,230 bp in all species. By comparing the LSC/IRb, SSC/IRa, and IRa/LSC regions, it was found that there were obvious differences in IRb/SSC regions

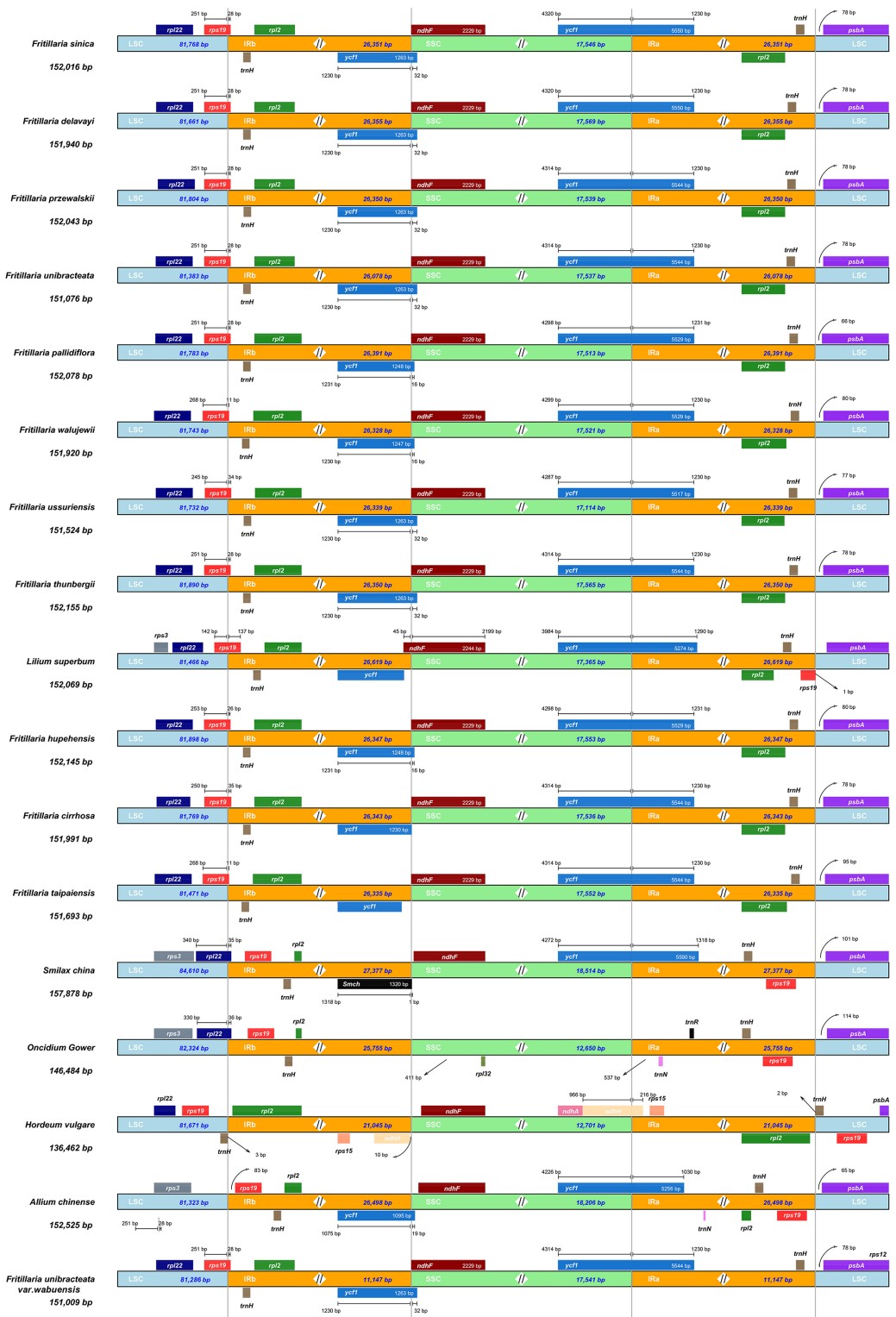

**Figure 2 Comparison of LSC, IRs, and SSC junction positions among 17 CP genomes.** Comparison of LSC, IRs, and SSC junction positions among 17 CP genomes.

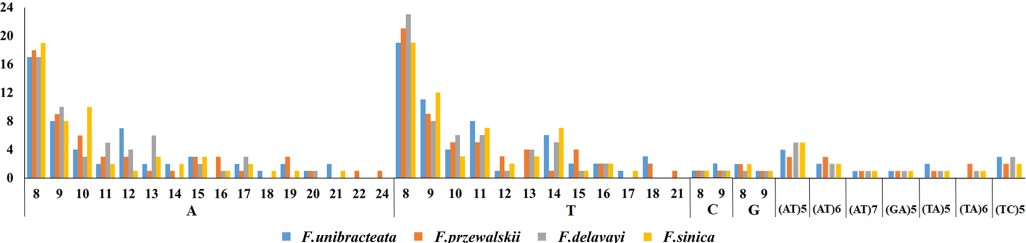

**Figure 3 Analysis of simple repetitive sequences in four *Fritillaria* CP genomes.** Analysis of simple repetitive sequences in four *Fritillaria* CP genomes     

among the 12 *Fritillaria* species. The *ycf1* gene in *F. taipaiensis* and *F. cirrhosa* did not cross the IRb/SSC boundary, whereas those in the other *Fritillaria* species extended 16 to 32 bp into the SSC region, which resulted a 16 to 32 bp overlap with *ndhF* gene.

## Repeat sequence, simple sequence repeats (SSRs), divergent regions and indels

The length of the repeat sequence ranged mainly from 15 to 20 bp and rarely from 21 to 38 bp among four *Fritillaria* species (Table S6). The repeating sequences were divided into forward repeating and palindrome sequences (including reverse and complementary sequences). The numbers of repeating sequences ranging from 15 to 20 bp of *F. unibracteata* and *F. przewalskii* were more than 487, while those of *F. delavayi* and *F. sinica* were less than 350 (Fig. S1). The numbers of repeat sequences in *F. przewalskii*, *F. unibracteata*, *F. sinica*, and *F. delavayi* were 1,200, 976, 656, and 425, respectively. Although repeat sequences with length ranging from 21 to 38 bp were rare, several promising molecular markers were found. For instance, *F. unibracteata* had three forward repeats at lengths of 23, 30, and 47 bp, respectively. *Fritillaria delavayi* also contained a palindromic repeat at a length of 54 bp, and *F. przewalskii* contained two forward repeats and a palindromic repeat at a length of 23 bp.

We also found 77, 76, 75, and 72 SSRs of at least 10 bp in *F. przewalskii*, *F. sinica*, *F. unibracteata*, and *F. delavayi*, respectively (Table S6, Fig. 3). These SSRs were mainly located in the LSC region, followed by 50 SSRs in IR region, and a few SSRs in the SSC region. The single- and three-nucleotide SSRs were the majority detected in these *Fritillaria* species, the double- and four-nucleotide SSRs were the minority detected, and a few were five-nucleotide SSR. Single- and three-nucleotide repeats in *F. unibracteata*, *F. przewalskii*, *F. delavayi*, and *F. sinica* together accounted for 81.33%, 83.12%, 79.17%, and 81.58% of SSRs, respectively. The single-nucleotide SSR with eight to nine repeated units were the most abundant and accounted for 53.91% of SSRs (Fig. 3). The high variation in numbers of SSRs might provide abundant information for molecular marker studies and plant breeding.

Using slide window analysis, 16 regions were eventually extracted to calculate the nucleotide variability with pi value ranging from 0.0104 (*rpl12*) to 0.0159 (*ycf1*). The top 10 most divergent regions were identified and thus might be used as potential molecular markers for future phylogenetic analysis and species identification in genus *Fritillaria*.

These regions included *ycf1*, *trnL*, *trnF*, *ndhD*, *trnN-trnR*, *trnE-trnT*, *trnN*, *psbM-trnD*, *atpI*, and *rps19* (Fig. S2). Due to its highest divergence, *ycf1* gene from 11 medicinal *Fritillaria* species was used to test its usefulness as a promising molecular marker for species identification. Based on our results, species-specific molecular markers for *F. ussurinensis*, *F. pallidiflora*, *F. taipaiensis*, *F. walujewii*, *F. thunbergii*, *F. hupehensis*, *F. unibracteata*, and *F. delavayi* were found (Fig. S3). *Fritillaria ussurinensis* (78 SNPs and nine indels) had the highest number of molecular markers, whereas *F. unibracteata* (two SNPs), *F. wabuensis* (two SNPs), and *F. delavayi* (two SNPs and two indels) had the least number of molecular markers. However, species-specific markers for *F. cirrhosa* and *F. przewalskii* were not found in the *ycf1* gene. For these two species, other highly divergent regions may provide better information for species-specific identification. Furthermore, due to its highest divergence, *ycf1* gene was used to construct phylogenetic tree in the following section.

Also, three indels, including a 137-bp deletion within *accD-psaI*, a 47-bp insertion within *trnG-GCC-trnR-UCU* and a 6-bp deletion within intron of *atpF*, were found in *F. taipaiensis*, *F. unibracteata*, and *F. cirrhosa* CP genome, respectively. Inspringly, these indels provided a basis of species-specific identification despite the fact that further experiments should be needed in the future.

## Phylogenetic tree on the basis of CP genomes

Prior to the phylogenetic analysis based on the CP genomes, we attempted to construct phylogenetic trees based on three common DNA barcodes from the CP genomes, including *matK*, *psbA-trnH*, and *rpl16*. Moreover, the *ycf1* gene was also used to construct the phylogenetic tree. As a result, *matK*, *psbA-trnH*, and *rpl16* obtained weakly supported trees, whereas phylogenetic tree based on *ycf1* gene was moderately supported as more than 50% and 60% branches got bootstrap values of more than 90 BP using NJ (Fig. S4) and ML methods (Fig. S5), respectively.

In addition, in comparison with four partial regions, the whole CP genomes obtained highly reliable phylogenetic tree. The CP genome matrix included the 27 *Fritillaria* species and 26 *Lilium* species, with three *Cardiocrinum* species and five *Amana* species as outgroups. On average, (152,099) bp of the CP genome were aligned. The result of ML tree was similar to that of NJ tree (Fig. S6). In the ML tree (Fig. 4), the ingroup corresponding to *Fritillaria* and *Lilium* was strongly supported (100 BP), and were sisters to *Cardiocrinum*. In this analysis, *Lilium* was monophyletic (100 BP) and was a sister to *Fritillaria*. Furthermore, *Lilium* nested with *Fritillaria* with moderate bootstrap support (75 BP) than that (53 BP) of previous report (*Day et al., 2014*). *Fritillaria*, as the largest subgenus, was paraphyletic and majority of which fell into one strongly supported Eurasain clade (A) except *F. maximowiczii* (subgenus *Liliorhiza*). Within the clade A, *F. davidii* appeared as successive sister taxa to the remaining Eurasian species (100 BP), which split into two well-supported clades. Clade A1 grouped with the monotypic subgenus *Rhinopetalum* (*F. karelinii*) as a sister to two species from subgenus *Fritillaria* (*F. ussuriensia* and *F. meleagroides*), which occurred in North region of China. The sister clade (A2) was composed of the remaining 22 species that could be classified into two

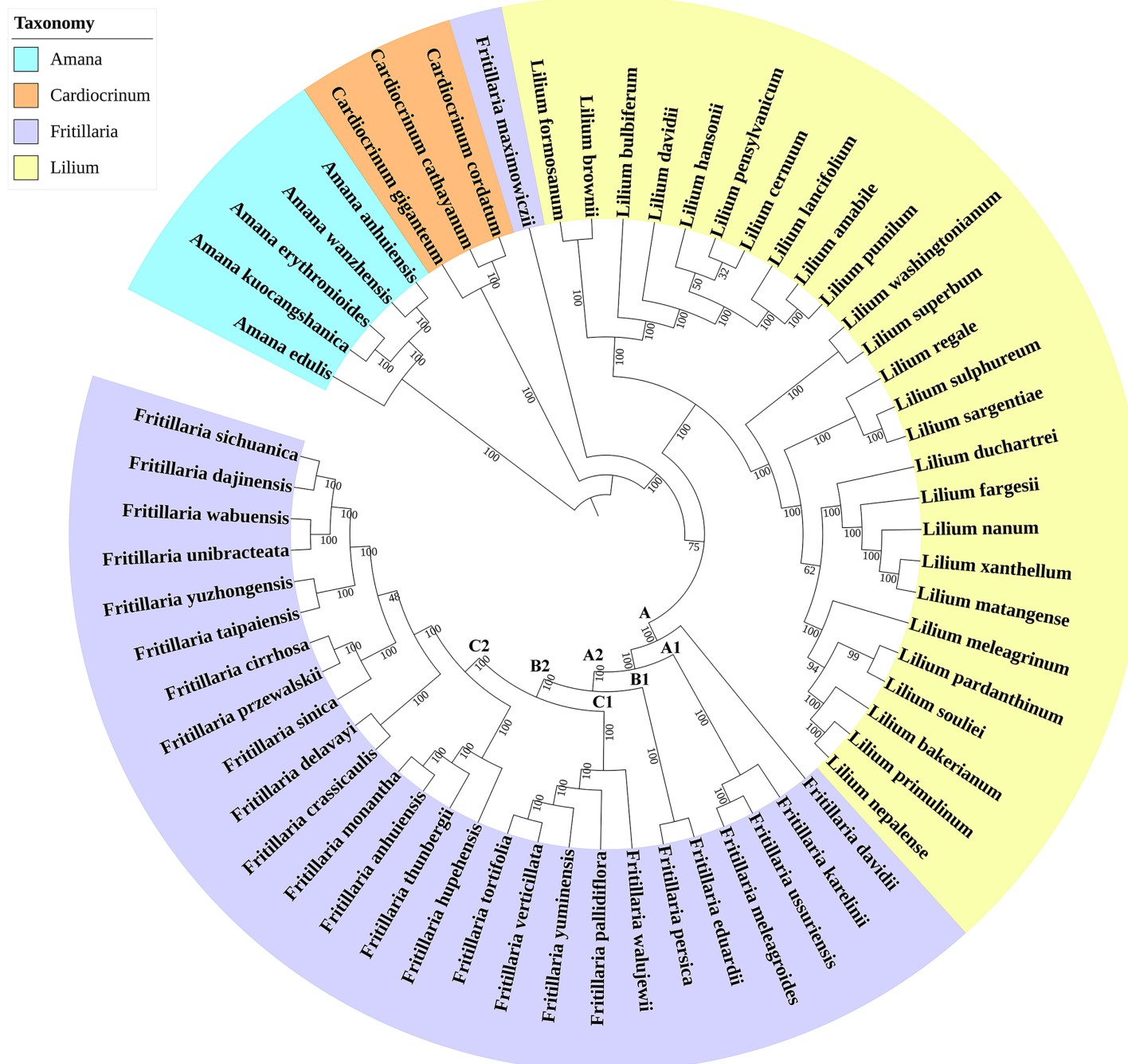

**Figure 4 Phylogenetic relationship of 61 species inferred from Maximum Likelihood tree.** Phylogenetic relationship of 61 species inferred from Maximum Likelihood tree. Numbers above nodes are supporting values with ML bootstrap values.

subclades (100 BP). Subclade B1 contained subgenus *Theresia* (*F. persica*) and subgenus *Petilium* (*F. eduardii*), which occurred in the Middle East and Central Asia, while subclade B2 comprising subgenus *Fritillaria* included 15 species from South China and five species (*F. tortifolia, F. verticillata, F. yuminensis, F. pallidiflora,* and *F. walujewii*) from Xinjiang Plain (Fig. 5). The 11 most valuable species used in traditional Chinese medicine

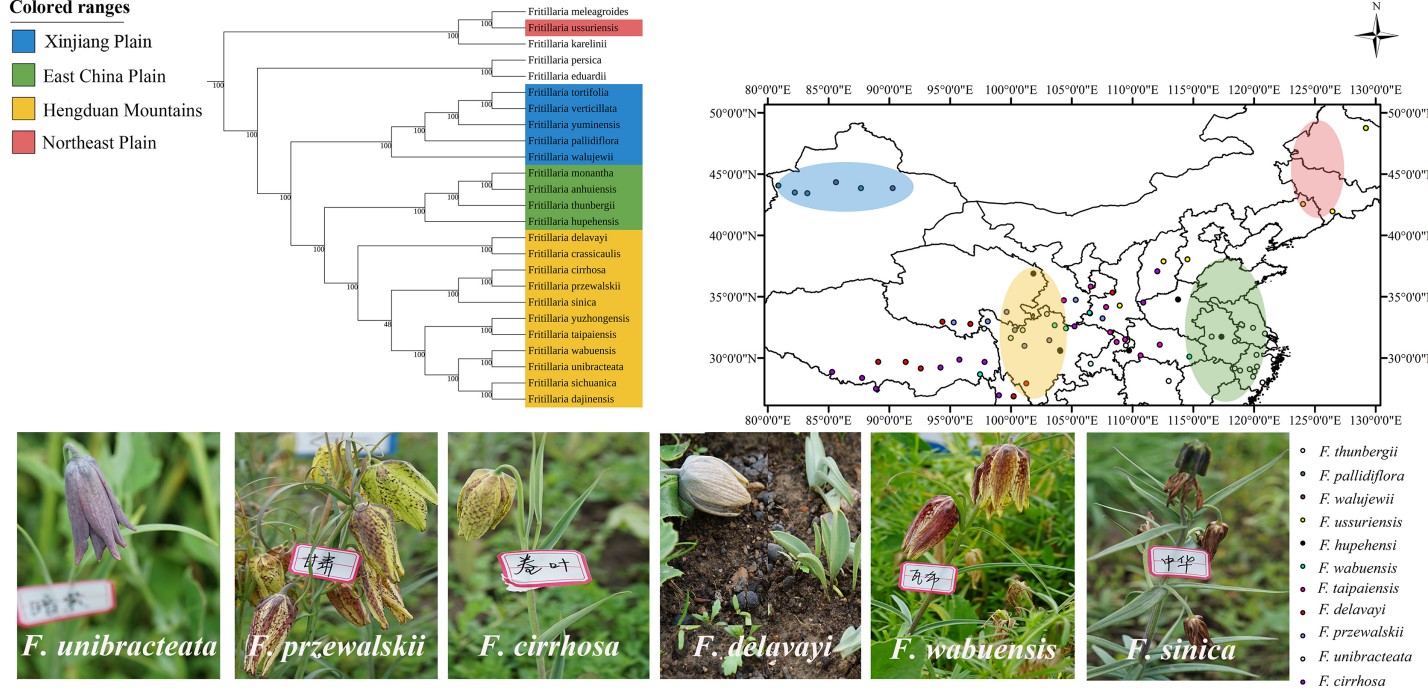

**Figure 5** **Distribution of 11 medicinal *Fritillaria* species in China.** Distribution of 11 medicinal *Fritillaria* species in China. The distribution area of each species is drawn according to the literatures and voucher specimens (http://www.cvh.ac.cn/). Photos of representative living plants of seven *Fritillaria* species Topographic data digital elevation modeling (DEM) data were required from the USGS website (https://glovis.usgs.gov/app?tour) with a 90-m spatial resolution grid.

were not in the same monophyletic group, as *F. ussuriensis* was separated from the other 10 species. As a whole, the phylogenetic tree based on the CP genome was highly supported, in which 91% (53 out of 58) branches obtained bootstrap values of more than 90 BP.

## DISCUSSION

### The overall structure of the CP genome

With the rapid development of *de novo* (Illumina) sequencing technology, the sequencing of CP genome is now cost-affordable and is much easier compared with the previous Sanger method. Moreover, *de novo* sequencing technology has been widely used in transcriptome assembly for identifying the biosynthetic and regulatory genes in traditional Chinese medicine, such as *Ligusticum chuanxiong* (*Song et al., 2015*) and *Cassia obtusifolia* (*Deng et al., 2018*). Here, four new CP genomes of *Fritillaria* were obtained using *de novo* sequencing technology. The CP genome sizes ranged from 151,076 to 152,043 bp, which were in accordance with those of reported CP genomes, such as *F. ussuriensis* (151,524 bp), *F. taipaiensis* (151,693 bp), and *F.cirrhosa* (151,991 bp). The four CP genomes contained similar genome structures, gene contents, and gene order, which were typical for land plants. Compared with the other three species, the number of *tRNA* and *rRNA* genes were the same, but the number of protein coding genes ranged from 77 to 78 due to the absence of the *rps16* gene in *F. unibracteata*. The absence of

*rps16* gene has also been observed in *Brassicaceae*, *Fabaceae*, and *Populus* species (*Jin, Choi & Choi, 2019*). The functional loss of the *rps16* gene from the CP genome could be compensated by the mitochondrial and (or) nuclear-encoded *rps16* gene that could target chloroplast as well as mitochondria (*Ueda et al., 2008*).

The highly conserved genomic structure and gene order as well as no rearrangement of the *Fritillaria* CP genomes have been observed in previous reports. The 26 kb of IRs in the *Fritillaria* species was within the size range of most angiosperm CP genomes (20 to 30 kb). The IR/LSC boundaries in the *Fritillaria* and *Lilium* (*Lilium superbum*) CP genomes expanded into the *rps19* gene, which might be a characteristic CP genome structure of *Fritillaria* and its relative genus. Similar expansion was also observed in other taxa from family *Liliaceae*, including *Lilium* (*Kim & Lee, 2004*), *Fritillaria* (*Li et al., 2014*), and *Cardiocrinum* (*Liu et al., 2018*). *Li et al. (2017)* reported that the common location of IR/LSC junctions in the *rps19* gene seemed to be an ancestral symplesiomorphy of *Liliaceae*. Here, similar feature was observed again, and the whole *rps19* gene was located inside the IR in *Smilax china*, *Oncidium gower*, and *Allium chinense*, while in *Hordeum vulgare*, the *rps19* gene did not extend into the IR (Fig. 2). The similar IRb/LSC boundaries among *Fritillaria*, *Lilium*, and *Cardiocrinum* implicated that these genera were closely related, which coincided with the phylogenetic result based on CP genomes (Fig. 4).

A careful comparison between repeat sequence and SSR regions revealed significant differences between various *Fritillaria* species, leading to the establishment of specific markers for molecular identification. In this study, a large number of repeat sequences, mainly with lengths ranging from 15 to 20 bp, were detected in the CP genomes of four *Fritillaria* species, consistent with the results on the CP genomes of *Cannabaceae* (*Zhang et al., 2018*). SSRs have been used for the study of population genetics because of their high variability (*Asaf et al., 2016*). The high ratio of SSRs in LSC region was also observed in *F. sichuanica* (*Chen, Wu & Zhang, 2019*). In the CP genomes of *F. unibracteata*, *F. przewalskii*, *F. delavayi*, and *F. sinica*, the contents of A/T repeats were much higher than those of G/C repeats, similar to the results of *Xue et al. (2019)* and other studies (*Melotto-Passarin et al., 2011*). Although several variable CP DNA markers, for instance, *matK*, *rpl16*, *atpB*, and *rbcL*, have been used in phylogenetic studies of *Fritillaria*, but they showed small divergence (pi value of 0.00717, 0.00571, 0.00391, and 0.00505, respectively) among the 12 *Fritillaria* species. Based on the result of sliding window analysis, the top ten divergent regions were identified with pi value ranging from 0.0116 to 0.0159. These highly divergent regions included *ycf1*, *trnL*, *trnF*, *ndhD*, *trnN*, *atpI*, and *rps19* in the coding region, and *trnN-trnR*, *trnE-trnT*, and *psbM-trnD* in intergenic region. The molecular markers found in the *ycf1* gene indicated that the highly divergent regions provided plentiful information for species-specific identification in the future (Fig. S3). Compared with the three common DNA barcodes, the *ycf1* gene generated a more reliable phylogenetic tree and it thus confirmed that highly divergent region was potential molecular markers for future phylogenetic analysis. The highly variable *trnE-trnT* and *ycf1* gene have also been reported by *Li et al. (2018)*, and the *ycf1*

gene has been proposed as the most promising plastid DNA barcode of land plants (*Dong et al., 2015*).

## The phylogenetic analysis of medicinal genus *Fritillaria*

Compared with partial sequences, the whole CP genome showed higher resolution with more than 91% branches having bootstrap value of 90 BP (Fig. 4). This result was consistent with a previous report (*Rønsted et al., 2005*) that increasing additional gene regions would help to improve the resolution. As suggested by *Kress et al. (2005)* and *Ng et al. (2017)*, the whole CP genome was promising as a super DNA barcode to resolve various *Fritillaria* species efficiently. Furthermore, our findings indicated that *Fritillaria* and *Lilium* were evidently sisters, with the closest relative being *Cardiocrinum* in a monophyletic genus (100 BP), similar with the result of *Chen, Wu & Zhang (2019)*. Such phylogenetic tree based on whole CP genome in this study showed similar topology with the previous study (*Rønsted et al., 2005*), but with higher resolution. Specially, genus *Fritillaria* was indicated as paraphyletic with higher bootstrap (100 BP) compared to 54 BP and 53 BP in the findings of *Rønsted et al. (2005)* and *Day et al. (2014)*, respectively.

The subgenus *Fritillaria* also appeared to be a paraphyletic group, similar to the results of *Day et al. (2014)*. One important medicinal *Fritillaria* species, *F. ussuriensis*, clustered with *F. meleagroides* and formed a sister clade to *F. karelinii* of subgenus *Rhinopetalum*, similar to the results of *Huang et al. (2020)*, *Khourang et al. (2014)*, and *Li et al. (2018)*. *Fritillaria ussuriensis* and *F. meleagroides* were frequently considered as members of the large subgenus *Fritillaria* (*Rix, 2001*). However, the two species do have some similarities with *F. karalinii* as both of them have small mastoid on filament, which are different from other species in Xinjiang Plain with no mastoid. Similar conflict between molecules and morphology was also observed in other taxa (*Anand et al., 2016*). Meanwhile, such mastoid on filament was proposed to be a potential primitive feature, and our results partly supported this hypothesis because *F. karelinii* and *F. meleagroides* diverged early from other medicinal species from Xinjiang Plain, such as *F. pallidiflora* and *F. walujewii*. Subgenus *Theresia* (*F. persica*) and *Petilium* (*F. eduardii*) had close relationship and formed monophyletic subclade B1, which was similar to the results of *Day et al. (2014)* and *Li et al. (2018)*.

As shown in Figs. 4 and 5, five species from Xinjiang Plain were included in a strongly supported subclade C1 (Fig. 4), which was a sister to subclade C2 containing the other 15 species from outside Xinjiang Plain. This signified that the Xinjiang species had a close genetic relationship. All the four species that were distributed in East China Plain, including *F. monantha*, *F. anhuiensis*, *F. thunbergii*, and *F. hupehensis*, nested in a supported subclade (100 BP). The remaining 11 species, including the complex group of *F. cirrhosa*, in another subclade were distributed in Hengduan Mountains (100 BP). The 11 important medicinal *Fritillaria* species were widely distributed in four hotspots, namely Xinjiang Plain, Northeastern China Plain, East China Plain, and Hengduan Mountains. The former two regions constituted hotspots in North China, while the latter two regions constituted hotspots in South China. Interestingly, the eight species in the upper location of the clade originated from Xinjiang Plain and Northeastern China Plain

(*F. ussuriensis*), whereas the 12 species in the lower location distributed in East China Plain and Hengduan Mountains region. Consequently, the geographical distribution pattern of the 11 medicinal species appeared to map on the phylogenetic tree, especially by plastid data (*Rønsted et al., 2005*). Similar result was also reported by *Li et al. (2018)*, and thus the investigation on the correlation between distribution pattern and phylogenetic relationship was needed in the future.

Early in 1987, *F. unibracteata*, *F. cirrhosa*, *F. prezewalskii*, and *F. delavayi* were recorded as national third-class endangered medicinal plants in China (*Konchar et al., 2011*). The most important medicinal species showed close relationship to widely cultivated members of subgenus *Fritillaria*, which raised the possibility of the rare species being replaced by those widely cultivated species. Recent analyses have demonstrated that *F. crassicaulis*, showing closest relationship with *F. cirrhosa*, has been widely used as the substitution of *F. cirrhosa* by people of Naxi nationality and Tibetan since Ming/Qing Dynasty (*Tang & Yue, 1992*). These findings highlighted those phylogenetic trees based on the CP genomes were promising in selecting potential novel medicinal species. In the future, those showing close relationship to the important species in traditional Chinese medicine, such as *F. sichuanica*, *F. dajinensis*, *F. yuzhongensis*, *F. sinica*, and *F. crassicaulis*, should be investigated to determine if these bulbs contain the same bioactive compounds found in the complex group of *F. cirrhosa*.

## The phylogenetic placement of some *Fritillaria* species

The non-monophyletic trait of subgenus *Fritillaria* indicated the incongruence in classification among some species, similar to the reports by *Rønsted et al. (2005)* and *Day et al. (2014)*. Although *F. ussurinensis* was regarded as a member of subgenus *Fritillaria*, its splitting from other members of subgenus *Fritillaria* has also been observed by *Chen, Wu & Zhang (2019)* and *Huang et al. (2020)*. There were several reports of natural interspecific hybrids (*e.g.*, *F. ussurinensis* (*Ruan et al., 2004*) and *F. eduardii* (*Wietsma et al., 2011*)), which might promote the molecular phylogenetic non-monophyly (*Funk & Omland, 2003*). Secondly, *F. davidii* had rice-shaped bulbils, resembling the morphological character of subgenus *Liliorhiza*, and used to be grouped in subgenus *Liliorhiza*. But based on our results, it was distantly related to subgenus *Liliorhiza* and was thus placed in subgenus *Davidii* as described by *Rix (2001)*. It was suggested that rice-shaped bulbils have independently evolved in *F. davidii* and subgenus *Liliorhiza* due to geographic separation, followed by a loss in some species in Eurasian clade during evolution (*Rønsted et al., 2005*). Thirdly, *Rønsted et al. (2005)* found that *F. pallidiflora* was resolved solely within the *Korolkowia/Petilium/Theresia* clade by combining plasmid *rpl16* and *matK* sequences. Our results demonstrated that *F. pallidiflora* clustered within subgenus *Fritillaria* and was more closely related to *Petilium/Theresia*. The conflict in *F. pallidiflora* was likely to be solved by using whole CP genome instead of separate regions. In addition, based on the CP genome, *F. unibraceata* was a sister to *F. wabuensis* with a divergence of 0.003, which was more than that between *F. sichuanica* and *F. dajinensis* (0.002). If *F. sichuanica* and *F. dajinensis* were given at rank of species, it was preferable to follow *Tang & Yue (1983)* and to rank *F. wabuiensis* as species instead of rank

of variant. However, this result was merely based on the CP genome, the accurate placement of *F. wabuensis* will be kept for further evaluation by nuclear genome comparison although it is extremely difficult to obtain.

## CONCLUSION

The CP genomes of the four *Fritillaria* species were useful resources for taxonomic clarification, determination of phylogenetic relationship and development of DNA markers. The phylogenetic tree based on the whole CP genome was reliable since 91% branches obtained bootstrap values of more than 90 BP, and the result supported the monophyly of genus *Lilium*, *Amana* and *Cardiocrinum*, except that the largest genus *Fritillaria* was paraphyletic. The 11 members of subgenus *Fritillaria* that were used in traditional Chinese medicine were split into two clusters since *F. ussuriensis* clustered with *F. meleagroides* and *F. karelinii*. In addition, the phylogenetic tree appeared to reflect a geographic distribution pattern of subgenus *Fritillaria*, and also highlighted the importance of the CP genome in the evolutionary analysis. The most important medicinal species, especially the complex group of *F. cirrhosa*, were found to be close to species that were in widespread cultivation for medicinal and ornamental purposes. Excitingly, those closely related species from subgenus *Fritillaria* might be promising alternatives to balance the improving market and rare resources. Finally, this study provided comprehensive molecular markers that might be valuable for future establishment of species-specific identification in *Fritillaria*.

### Availability of data and materials

The chloroplast genomes generated during the current study were deposited in NCBI with accession number of MW849272 (*F. unibraceata*), MW849274 (*F. przewalskii*), MW849275 (*F. delavayi*) and MW849273 (*F. sinica*), respectively. All the raw Illumina data of *F. unibracteata*, *F. przewalskii*, *F. delavayi* and *F. sinica* have been deposited in the Sequence Read Archive (SRA) of the NCBI under accession numbers of SRR14454932, SRR14455034, SRR14454929 and SRR14455331, respectively.

### Funding

This work was co-supported by the National Natural Science Foundation of China (No. 31500276), the Sichuan Science and Technology Program (No. 2018SZ0061 and 2021ZHFP0170), the Sichuan Administration of TCM program (No. 2021MS116), and the Fundamental Research Funds for the Central Universities (No. 2682021ZTPY017). The funders had no role in study design, data collection and analysis, decision to publish, or preparation of the manuscript.

### Grant Disclosures

The following grant information was disclosed by the authors:
National Natural Science Foundation of China: 31500276.

Sichuan Science and Technology Program: 2018SZ0061 and 2021ZHFP0170.
Sichuan Administration of TCM program: 2021MS116.
Central Universities: 2682021ZTPY017.

## Competing Interests

Tiechui Yang is an employee of Qinghai lvkang Biological Development Co., Ltd.

## Author Contributions

- Tian Zhang performed the experiments, analyzed the data, prepared figures and/or tables, and approved the final draft.
- Sipei Huang performed the experiments, analyzed the data, prepared figures and/or tables, and approved the final draft.
- Simin Song analyzed the data, prepared figures and/or tables, and approved the final draft.
- Meng Zou analyzed the data, prepared figures and/or tables, and approved the final draft.
- Tiechui Yang analyzed the data, prepared figures and/or tables, and approved the final draft.
- Weiwei Wang performed the experiments, prepared figures and/or tables, and approved the final draft.
- Jiayu Zhou conceived and designed the experiments, analyzed the data, authored or reviewed drafts of the paper, and approved the final draft.
- Hai Liao conceived and designed the experiments, analyzed the data, authored or reviewed drafts of the paper, and approved the final draft.

## DNA Deposition

The following information was supplied regarding the deposition of DNA sequences:

The chloroplast genomes generated during the current study are available at NCBI: MW849272 (F. unibraceata), MW849274 (*F. przewalskii*), MW849275 (*F. delavayi*) and MW849273 (*F. sinica*), respectively.

Data can also be found at:

DOI 10.6084/m9.figshare.14531937.

DOI 10.6084/m9.figshare.14531946.

DOI 10.6084/m9.figshare.14531961.

DOI 10.6084/m9.figshare.14531970.

## Data Availability

Data is available at NCBI: MW849272, MW849274, MW849275 and MW849273.

The data will be relased by NCBI after the reviewing process. Please see the "Proof of accession number" in Supplemental File.

Moreover, all the raw Illumina data of *F. unibracteata*, *F. przewalskii*, *F. delavayi* and *F. sinica* are available at the Sequence Read Archive (SRA) of the National Center for

Biotechnology Information (NCBI): SRR14454932, SRR14455034, SRR14454929 and SRR14455331.

## Supplemental Information

Supplemental information for this article can be found online at http://dx.doi.org/10.7717/peerj.12612#supplemental-information.

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
