# Peer review of "Identification of evolutionary relationships and DNA markers in the medicinally important genus Fritillaria based on chloroplast genomics"

_PeerJ, doi:10.7717/peerj.12612_

## Round 0.1 · original submission · Major Revisions

I have now received two reviewer reports for your study, both of which are extensive and constructive in nature that will help enhance your study. They both feel the study is useful, the data is extensive but feel that the presentation could greatly be improved. Please address all of their comments and provide a point by point rebuttal.

Especially pay attention to the following.

1. Provide a clear and concise discussion on the relevance of your results. The discussion as it is is meandering and too long.

2. Improve the clarity, reduce verbosity and address the syntactical issues pertaining to language.

3. Improve on the explaining of the methodology as the reviewers also suggest (both reviewers make suggestions on how to improve on this front)

4. Be cautious on interpretation of the results based on poorly supported clades of the phylogeny

·

Basic reporting

This is a well-written manuscript that uses clear, professional English throughout. The literature references were appropriate and adhered to the format of the journal.

Major comments

The manuscript is laden with redundant information. This makes it quite difficult to follow the main narrative and sieve out the important information. The length of the manuscript can be substantially reduced. Suggested improvements include:
1. Remove redundant background information from the Introduction (see annotated PDF).
2. Move at least half of the table/figure to the Appendix (see annotated PDF).
3. Provide succinct summary of the Results in the main text and avoid repeating information (especially numbers) that are already included in tables/figures (see annotated PDF).

Related to the point above, the study includes several analyses that did not directly address the research questions posed in the Introduction. The connection between these analyses and the research questions should be made clearer. They include:
1. RSCU and codon usage bias analysis
2. InDel identification
3. Divergence hotspot identification

Also, one specific section on the "Specie-specific test for F. taipaiensis, F. unibracteata and F. cirrhosa" appeared abruptly in the Results section, without background information or explanation of its relevance in the Introduction and description of the analysis in the Methods. This section should be better incorporated with the rest of the manuscript. In the Discussion, this section appeared out of place and the findings were too preliminary to justify its reporting in the manuscript. Perhaps it should be tested against all other Fritillaria species?

Finally, the authors should make a clearer case on why the additional cpgenomes of F. unibracteata, F. przewalskii, F. delavayi and F. sinica are necessary for this study. These four species were included in the largest clade in Fritillaria and may not be crucial for examining the general phylogeny of Fritillaria and Liliaceae.

Experimental design

The knowledge gaps and research questions are well defined and have broad relevance to plant science. The methods are clearly described and the experimental design is appropriate and rigorous.

Validity of the findings

No comment.

Reviewer 2 ·

Basic reporting

Clear, unambiguous, professional English language used throughout.

No. Numerous spelling, formatting and grammar issues are apparent throughout the text. Also, occasional use of non-professional or subjective language is present. Technical terms (specie-specific, subgenera when singular, conservative instead of conservation etc) need to be thoroughly checked throughout the manuscript. The use of a professional language service or a native English speaker is required to thoroughly check the text. Duplicated spacing, punctuation and tabs occur throughout the text and need to be removed.

Intro & background to show context. Literature well referenced & relevant.

Authors provide decent level of background information but also introduce non-scientific information which is presented as scientific literature (e.g. ‘medical’ terms, symptoms) and TCM related items which are not known/used outside China. Non-relevant information is particularly present in the introduction and needs to be removed to bring forward to reason and goals of the study more clearly. Key literature is missing form the introduction, especially of authors having done very similar studies recently (see below).

Morphological traits mentioned are not characters which are known as particularly diagnostic, and hence provide a very shaky foundation to base a classification on. The term ‘ecological stress factors’ is unclear – natural ecological variation is more likely to be appropriate here but the authors do not provide enough information to be sure.

Structure conforms to PeerJ standards, discipline norm, or improved for clarity.

Yes.

Figures are relevant, high quality, well labelled & described.

Yes, but some figures appear to be duplicated in the supplementary materials. I feel many of the Figures could actually be better placed in the Supplementary figures.

Raw data supplied (see PeerJ policy).

Yes.

Experimental design

Original primary research within Scope of the journal.

Yes, somewhat. Part of the work presented has also been done previously by other authors, not all of whom have been cited by the authors.

Research question well defined, relevant & meaningful. It is stated how the research fills an identified knowledge gap.

Yes.

Rigorous investigation performed to a high technical & ethical standard.

High technical: No.
Ethical: yes.

Methods described with sufficient detail & information to replicate.

Yes.

Validity of the findings

Impact and novelty not assessed. Meaningful replication encouraged where rationale & benefit to literature is clearly stated.

Setting aside any assessment of impact and novelty, I feel the study is not well written out and the results are presented in a way that inflates the actual value. The setup is not thought through well, giving cause to unfounded speculation in the discussion.

All underlying data have been provided; they are robust, statistically sound, & controlled.

Yes. No, No, No. (see comments above and below)

Speculation is welcome, but should be identified as such.

The discussion is far too elaborate, and inflating text is used to expand results beyond its actual value. Many off shoots are described, distracting from the actual results. The authors should refocus the Discussion by sticking to the data generated here and its immediate impact/promise. In particular the phylogenetic results need to be re-assessed, using the scale provided above. A substantial part of the branching is not or weakly supported. Overall, I feel the authors do not have a solid grasp of how to evaluate phylogenetic results and their implication on taxonomy, ecology and a classification in general.

Conclusions are well stated, linked to original research question & limited to supporting results.

No.

Additional comments

Line 54-58: remove subjective language and unnecessary text.

Line 68: remove the word ‘clinical’

Line 69-71: use scientific language for ailments

Line 90: ‘since the bulbs of some Fritillaria species showed great economic value in Asian countries’ – this needs a reference and a value, it is not listed elsewhere, except for China

Line 96: ‘ Identifying the closest relatives could point to additional species that might be analyzed for their potential medicinal value, which might in turn reduce pressure on those species that were currently faced with survival risk.’
Will this not lead to overharvesting there? This is not a sustainable research approach as suggested in the abstract.

Line 98: ‘genuine medicinal plants’ – what is meant by this? This seems highly subjective and contentious as many ‘medicinal’ uses have not been clinically proven.

Line 112: There is no such thing as a ‘partial phylogenetic signal’. It is either present or not, and then it can be weak or strong. Rephrase throughout.

Line 150: you can not assess the utility of molecular markers for population genetics in a study employing only 1 or a few individuals for each species. Remove.

Line 180, 182, 183, 190, 191, 303, : reference missing

Line 201: indels. Correct unnecessary capitalization throughout.

Line 219: Table 1 not relevant, can be summarized in the text or moved to the Supplementary Materials.

Line 257: where is the statistical proof for this statement of significance? If performed, show here, or do a statistical test and show the evidence, or change the statement.

Line 319: unclear, what is the difference between these groups.

Line 325: Vague statement – quantify, and confusing reasoning.

Line 325-346 Strong support >90, Moderate support 80-90, Weak support 65-80, No support <65. Be critical of the results! Many of the discussed nodes have no support whatsoever.

Line 368, 445-467: reassess statements with above valuation of support.

Line 375-385: I have considerable difficulties with this method of assessing species-specific differences, based on only one or several individuals in the study. This is not scientifically verifiable at all, unless more individuals for each species are added. The differences at times are so minute (a 6 bp deletion is reported), that this begs the question how prevalent this is in the wider population of this particular species. Without multiple individuals, the claim that this ‘species-specififc’ is impossible to evaluate, especially in a group where species distinction using morphological characters is difficult and the position of cryptic species complicates identification.

Line 390-393: confusing and not relevant to the manuscript. This is a cp paper and sequencing a whole nuclear genome is out of the original scope and aims, requiring a substantially different approach not employed here.

Line 396, 421: provide correct taxonomic names.

Line 422-424: confusing. Rephrase.

Line 441: What is ‘Internal Gene Space’?

Line 568, 571: Removing subjective text.

Line 388-617: the Discussion is much too long and introduces far too many aspects which are beyond the scope of this study. It needs to be fully rewritten and extensively cleansed of topics not relevant to the experiments performed by the authors.

Line 620: reassess as stated above.

Line 623-626: For this to be valid, this needs to be looked at in much greater detail. At current, there is no correlation provided between geography, traits and phylogeny. At best, one could say that the tree appears to reflect a geographic distribution pattern.

Line 629-631: doubtful and hard to substantiate, without the addition of more individuals for each species.

---

## Round 0.2 · Major Revisions

I have received two reports for your paper. Both reviewers agree that the paper has been substantially improved, but recommend improvements in writing/clarity. I went through the manuscript up to the introduction section and point out many errors in writing (please see attached PDF). Hence please revise the entire manuscript focusing on grammar and readability. One reviewer also brings up two important issues pertaining to experimental design and justification of the study. Please address these issues also and resubmit a revised version of your manuscript.

·

Basic reporting

English
* * *
Upon closer reading, it was apparent that substantial improvement on the writing is necessary. There are many grammatical/spelling errors and incorrect terminology; some examples are provided below.

Line 66-67. "Nowadays, DNA-based classification occurred has been applied in angiosperm phylogeny group because of their reliability and readable data" should be "Molecular systematics has been widely applied to clarify angiosperm phylogeny".

Line 67-68. "Basically, accurate identification (eg. using DNA markers) was necessary to discriminate among the original specie and its adulterants" Do you mean "cryptic species"? "Adulterants" usually refer to substance/chemical that is used to alter properties of food product.

Line 70-71. "...the wild Fritillaria populations decreased sharply due to long-term excessive harvesting" should be "...the wild Fritillaria populations experienced sharp decline due to long-term overharvesting".

It is important that the manuscript goes through English editing before it can be ready for publication.

Literature, background and article structure
* * *
The manuscript adhered to professional article structure and covered sufficient literature and background.


Relevant results to hypotheses
* * *
Yes, the results are relevant to the research question.

Experimental design

Knowledge gaps and research questions
* * *
I appreciate the authors' careful response to my previous comments. The current version of the manuscript is more succinct than before, especially with the removal of redundant information.

However, I am unconvinced that there is a need to perform de novo assembly for the cpgenome of F. unibracteata, F. przewalskii, and F. sinica again, given that this had been done in Chen et al. (2019). Why wasn't the previous cpgenomes used in the study here? What is the value of assembling the cpgenome, de novo, again?

Also, since one of the main objectives was to obtain candidate DNA markers that could potentially be used in species identification of medicinal Fritillaria, the study seemed incomplete with just indel markers that can identify three out of 11 medicinal species. I strongly suggest removing this component. Alternatively, the authors should explore/report markers that can be useful in identifying all 11 medicinal species. How about the DNA barcoding regions? Or ycf that is highly divergent? What would the authors suggest future studies explore in order to find markers for quick and easy species identification?

These would have to be addressed to clarify how the study address existing knowledge gaps.

Validity of the findings

All good. No comment.

Reviewer 2 ·

Basic reporting

Following the initial round of reviews and comments provided, I can see that the authors have performed an extensive round of revision and addressed most of the issues highlighted before.

The use of English and correct spelling/grammar has been very much improved. Background research and citations have been improved. The article overall reads much better as a large proportion of redundant text has been removed.

Experimental design

The used methodology is better explained.

Validity of the findings

Interpretation and description of results has been very much improved.

Additional comments

As a minor formatting comment, I would suggest to the authors to read through the manuscript one more time in detail, paying particular attention to the proper use of singular and plurals (and correct formulation of sentences as such), and the unnecessary use of capital letters. Also, I still find four more usages of the word 'specie'. The manuscript has been very much improved from the initial submission overall.

---

## Round 0.3 · Minor Revisions

Now I have received the reviewer report to your study. You have addressed most of the comments of the reviewer to an appreciable degree, the paper is much improved, and is almost ready to be accepted. However, as the reviewer points out, the language is not polished enough for the paper to be published as-is. So I request the authors to improve on this front, especially the discussion.

·

Basic reporting

I thank the authors for engaging a native speaker to edit the language of the revised manuscript. The content of the manuscript is almost ready, but I do think that it still needs a final, professional English editing (perhaps from the journal?) to be publishable. Case in point, the first sentence of the abstract is missing an article "the".

Other examples of improvements needed on language editing include:

Line 243-245: As a consequence, ycf1 gene as well as other highly divergent regions would provide useful information for species-specific identification in the future.
Suggestion: For these two species, other highly divergent regions may provide better information for species-specific identification."

Line 403-404: The CP genomes of four Fritillaria species provided support for taxonomic clarification, phylogenetic relationship and development of DNA markers.
Suggestion: The CP genomes of the four Fritillaria species were useful resources for taxonomic clarification, phylogenetic relationship and development of DNA markers.

Experimental design

It is greatly appreciated that the author provided further background on why the cpgenomes of F. unibracteata, F. przewalskii, and F. sinica were assembled, de novo, in this study. Genomic resources are being rapidly produced, so it is completely understandable that other groups concurrently developed these cpgenomes. This is valuable information to the reader and I am glad that author added the explanation.

The exclusion of the preliminary ITS tests and in its place, the inclusion of a more comprehensive and applicable marker development (involving ycf1 and other regions) is highly welcomed. It provides an important resource for species identification, which has wide applications in TCM and species conservation. The new tests are thorough and reliable.

With these two major edits, the study design is now more concise and clearer.

Validity of the findings

No comment.

---

## Round 0.4 · accepted · Accept

This paper has now been revised for technical issues, clarity and language to an appreciable degree and hence reaches the editorial criteria for acceptance. I congratulate the authors on this contribution and wish them the best with their future research work.